# New Challenges and Prospective Applications of Three-Dimensional Bioactive Polymeric Hydrogels in Oral and Craniofacial Tissue Engineering: A Narrative Review

**DOI:** 10.3390/ph16050702

**Published:** 2023-05-05

**Authors:** Gamal Abdel Nasser Atia, Hany K. Shalaby, Naema Goda Ali, Shaimaa Mohammed Morsy, Mohamed Mohamady Ghobashy, Hager Abdel Nasser Attia, Paritosh Barai, Norhan Nady, Ahmad S. Kodous, Hasi Rani Barai

**Affiliations:** 1Department of Oral Medicine, Periodontology, and Diagnosis, Faculty of Dentistry, Suez Canal University, Ismailia P.O. Box 41522, Egypt; 2Department of Oral Medicine, Periodontology and Oral Diagnosis, Faculty of Dentistry, Suez University, Suez P.O. Box 43512, Egypt; 3Radiation Research of Polymer Chemistry Department, National Center for Radiation Research and Technology (NCRRT), Egyptian Atomic Energy Authority, Cairo P.O. Box 13759, Egypt; 4Department of Molecular Biology and Chemistry, Faculty of Science, Alexandria University, Alexandria P.O. Box 21526, Egypt; 5Department of Biochemistry and Molecular Biology, Primeasia University, Dhaka 1213, Bangladesh; 6Polymeric Materials Research Department, Advanced Technology and New Materials Research Institute (ATNMRI), City of Scientific Research and Technological Applications (SRTA-City), New Borg Elarab, Alexandria P.O. Box 21934, Egypt; 7Department of Radiation Biology, National Center for Radiation Research and Technology, Egyptian Atomic Energy Authority (EAEA), Cairo P.O. Box 13759, Egypt; 8Department of Mechanical Engineering, Yeungnam University, Gyeongsan 38541, Republic of Korea

**Keywords:** three dimensional, hydrogel scaffolds, tissue engineering, polymeric hydrogels and tissues regeneration

## Abstract

Regenerative medicine, and dentistry offers enormous potential for enhancing treatment results and has been fueled by bioengineering breakthroughs over the previous few decades. Bioengineered tissues and constructing functional structures capable of healing, maintaining, and regenerating damaged tissues and organs have had a broad influence on medicine and dentistry. Approaches for combining bioinspired materials, cells, and therapeutic chemicals are critical in stimulating tissue regeneration or as medicinal systems. Because of its capacity to maintain an unique 3D form, offer physical stability for the cells in produced tissues, and replicate the native tissues, hydrogels have been utilized as one of the most frequent tissue engineering scaffolds during the last twenty years. Hydrogels’ high water content can provide an excellent conditions for cell viability as well as an architecture that mimics real tissues, bone, and cartilage. Hydrogels have been used to enable cell immobilization and growth factor application. This paper summarizes the features, structure, synthesis and production methods, uses, new challenges, and future prospects of bioactive polymeric hydrogels in dental and osseous tissue engineering of clinical, exploring, systematical and scientific applications.

## 1. Introduction

Tissue and organ deficiencies have been highlighted as a significant public health concern, with only a tiny number of qualified individuals obtaining transplants [1]. The majority of tissue and organ waiting lists do not correctly represent the severity of the situation because only the sick people need this type of medical attention [2]. In this article, the terms “regenerative medicine” and “tissue engineering” are used interchangeably by researchers and medical professionals [3].

The capacity and capability to repair and restore injured organs and tissues underlies the promises of regenerative medicine [4]. Regenerative medicine has demonstrated promise outcomes for the regeneration and substitution of a vast range of tissues and organs, including bone, cartilage, teeth, kidney, and liver, as well as the possibility to cure some developmental problems [5]. The conventional dependence on donated tissues and organs for implantation risks donor limitations as well as the possibility of immunologic resistance of the delivered body part [6]. Some organ transplants conducted in impoverished countries involve transplant tourism, in which tourists with adequate money and power are given preference over the local population [7]. Such tactics have been denounced since they can lead to the abuse of vulnerable persons [8]. Regarding disparities in regional economic strength, and hence inequalities in healthcare system, eliminating obstacles such as the scarcity of organs and the practical challenges of obtaining and keeping them might assist in increasing the number of individuals who can get organ transplants [9]. As a result, techniques and technology to expand the supply of tissues and organs for transplantation should be further improved.

In most circumstances, tissues and organs are needed for transplantation, such as when individuals are injured in accidents, conflicts, or natural catastrophes [10]. The scarcity of tissues and organs not only impedes medical care but also scientific study.

The production of an unlimited number of tissues and organs is thus the most difficult challenge of our age. Many campaigns have been launched in order to encourage organ donations and improve the utilization of donor organs [11]. Regenerative dentistry and regenerative medicine are hot topics because the majority of people needs tissue regeneration because of increased debilitating diseases and disorders [12]. Furthermore, methods to minimize postsurgical complications must be considered for people. Regenerative dentistry and medicine are interdisciplinary fields that include stem cell technologies, tissue engineering, and material science.

They use physiological functions to rejuvenate and rebuild tissues’ functionality [13]. Despite developments in the therapeutic use of scaffolds for dental and osseous regeneration, the most reliable techniques of filling defects with calcified tissue walls to promote tissue healing remain the most predictable [14].

Tissue engineering is a viable alternative for regenerating damaged osseous tissues and restoring their biological properties; several studies have proved the usefulness of this technique for regenerating orofacial bones [15]. Tissue engineering techniques generally consist of three important components: a stem cell compartment, signaling substances (e.g., growth factors), and scaffold materials.

Scaffold materials should be osteoconductors, promoting bone-forming cells and preserving neo-vascularization that will provide nutrition to the new cellular environment, and osteoinducers, delivering biologically active compounds gradually that induce stem cells to specialize and accumulate calcified tissues [14]. Furthermore, the tightly controlled structure of osteons in the natural bone architecture shows that cell quantity, source, and gap are crucial for scaffolding architecture and fabrication. Figure 1 shows the macroscopic and microscopic structure of bone [16].

Hydrogels are cross-linked polymers of hydrophilic groups. The strong aqueous capacity allows it to absorb enormous volumes of water or liquids while staying basically insoluble. Furthermore, hydrogels integrate well with the surrounding structures, limiting the potential of inflammatory reactions [17]. Natural to artificial hydrogels are accessible in a variety of forms, boosting their flexibility.

Hydrogel-derived biomaterials have the potential to improve existing therapeutic methods to bone tissue creation. The framework has to possess sufficient bioactivity, and biodegradability, with adequate architecture that resembles environment (niche) to allow cell attachment, multiplication, growth, and de-novo tissue formation [18]. A scaffold can be classified as pre-formed or injectable based on when it is formed.

A pre-adjusted framework has a predetermined shape before being applied, in contrast, an injectable matrix takes the shape of the deformity. Polymeric scaffold has several advantages over the pre-formed scaffold, such as: (1) It is carried out in a painless fashion, lowering the risk of contamination and enhancing satisfaction; (2) it can simply accommodate any unorganized deformities; and (3) it triumphs over the problems of cell attachment, as these considerations can be primarily combined with the material solution prior to getting inserted in situ. Because of the volume, form, and complicated architecture of dentofacial tissues, an injectable biomaterial is optimal [19].

This is a narrative review that addresses the current status of the knowledge of polymeric hydrogels for oral, and craniofacial tissue regeneration from both a conceptual and practical standpoint.

## 2. Bio Scaffolds for Dental and Osseous Tissue Engineering

The approach of biomaterial-dependent tissue engineering was established in the early 1990s to address the limited therapeutic effectiveness of cells implanted into the defected region, because of poor cell retention at the wounded site. There are various drawbacks to using grafts generate bio-active replacements that recover, preserve, or increase cellular activity, including a lack of tissue regeneration, an unsatisfactory delivery of nutrients, and inadequate metabolic end elimination [20]. As a result, it is vital to establish a sufficient milieu for cell viability and functionality, which might be accomplished with 3D designed biomaterials, such as scaffolds [21]. For several years, bone auto grafts were thought to be the golden therapeutic measure for bone regeneration; however, their clinical uses are restricted due to scarcity and, more crucially, donor-site complications [22]. Allografts are simple to obtain and handle, however they may induce hypersensitivity [23]. Consequently, there is an increased demand for innovation of artificial bone grafting substances that may effectively replace and enhance bone tissue.

Several biomaterials, including bio ceramics [24], metals [25], and polymers [26] have been investigated for tissue regenerative strategies, as shown in Table 1.

Hydrogels may offer numerous benefits in tissue repair. They are polymeric frameworks made up of cross-linked hydrophilic domains that provide mechanical stability and nutritional conditions conducive to endogenous cell development.

## 3. Pre-Requisites of Hydrogels for Dental and Osseous Regeneration

Hydrogels have gained popularity in medical applications, because they are biocompatible, biodegradable, resemble the ECM (extracellular matrix) architecture, and easy to use in practice [27]. Given the advancement of personalized medicine, there is a growing necessity to produce hydrogels to transport biological substances, eliminating the need for open surgery and enabling from minimally invasive techniques to biomaterial and cell application [28]. Scaffolds should be economic to allow their massive clinical application. Figure 2 shows the requirements of tissue engineering scaffold [29].

### 3.1. Cytotoxicity

Before, during, and after injection, the injectable hydrogels should be non-toxic, with gentle solidification conditions. The scaffolds used to enclose cells must be gel-able without harming the cells. Injectable materials must not contain any initiators, cross linkers, or other compounds that are harmful to the cells. The polymeric process should be carried out without raising the temperature to the point where thermal damage occurs at the defect site. There should be no immunologic responses in the scaffolds. Naturally originated hydrogels are generally regarded safe; however, the chemicals utilized as cross linkers could be harmful to some level. Nevertheless, cytotoxicity is a serious problem for artificial hydrogels [30]. To now, the FDA (Food and Drug Administration) has only authorized PLA (polylactic acid—a biopolymer), PEG, and PLGA for clinical purposes. Other gelation process components, such as promoters, chemical agents, stabilizers, and moisturizing ingredients, may be cytotoxic if not properly eliminated or if they penetrate into tissues or entrapped cells.

### 3.2. Biological Responses

The host reaction to a scaffold is critical, which implies that the biomaterial must not evoke a persistent inflammatory process or exhibit severe cytotoxicity [30]. A biocompatible substance would promote cell development and multiplication with absence of toxicity or an immune reaction. The level of cell attachment on biomaterials is a crucial indicator of biocompatibility since it is required for both cell adherence, and up-regulation of several downstream processes that govern cellular activity. Because most natural polymers feature structural units that are comparable to physiologic ECM and the cell-adherence domains are often kept during the creation process, natural polymer hydrogels are widely considered biocompatible materials. Nevertheless, because most natural hydrogels are generated from animals, caution must be used when using them in tissue engineering to limit the danger of cross-infections, and inflammatory responses. Because synthetic hydrogels frequently lack cell attachment sites, changes are generally required to generate enhanced host responsiveness. Ideally, RGD (Arg-Gly-Asp) peptide insertion is beneficial for artificial biomaterials, whereas additional changes such as the introduction of fibronectin sequences as CD44 and CD168 are also required [31]. Furthermore, copolymer hydrogels derived from natural and artificial polymers have improved the biocompatibility.

### 3.3. Mechanical Characteristics

Another key aspect for biomaterials used as frameworks in tissue engineering is optimal structural integrity. Following processing, hydrogels should have appropriate mechanical strength. In situ gelation is needed to endure biomechanical loads and offer transient assistance for the cells [32]. The structural integrity of the biomaterial is primarily determined by the original stiffness of the polymer subunits, the kinds of cross linker, the density of the crosslinking, and shrinkage as a consequence of the hydrophilic/hydrophobic equilibrium [33]. Because of their thermodynamic compatibility with water, injectable hydrogels are durable in comparison to other scaffolds.

Hydrogels should sustain the cells and endure mechanical stress, they must have sufficient mechanical characteristics [34,35]. Along with crosslinking density, permeability, and hydrogel adaptation, hydrogel structure, quantity, and production process also have a significant role in hydrogel mechanical characteristics [36,37,38]. By reducing the material’s porosity and increasing crosslinking density, which can impair cellular responsiveness and degradability, the mechanical characteristics of the material can be improved [34,39]. Thus, there should be an equilibrium between the mechanical characteristics of hydrogels and their capacity to degrade [40].

The introduction of microspheres, or biocermics into the polymer improves the mechanical characteristics of injectable scaffolds [41]. The rigidity and hardness of injectable hydrogels are two essential mechanical properties. A hydrogel should be stiff enough to create a somewhat stable framework for cells and strong enough to protect the framework from premature degradation. Increasing the intensity of crosslinking enhances the firmness and durability of the hydrogel. The deterioration period of the scaffolding with increased crosslinking capacity, on the other hand, is likewise prolonged. As a result, for the implementation of hydrogels, a balance between structural integrity and degradation is required.

### 3.4. Degradation

The materials employed to create hydrogels must gradually degrade and make room for freshly created tissues. The breakdown speed is preferably to be equal to the pace of new tissue synthesis. Hydrogels can be decomposed by one of three ways: simple dissolving, hydrolysis, or enzymatic breakage. Hydrogel generated by physical cross linking cannot preserve their structure and dissolve in a solution when environmental factors such as temperature and pH vary. Ionic cross-linked hydrogels react to changes in ion strength in solution, and the hydrogel dissolves as the ions leak out of it. Because the hydrogel components are not broken down into little molecules, simple dissolving does not result in degradation. The most typical process of hydrogel breakdown is degradation of labile ester bonds in the matrix core (e.g., lactide or glycolide fragments in PLGA) [42]. The hydrolysis rate is determined by factors such as cross-linker concentration, molecular mass, shape, porosity, and the quantity of remaining monomer.

Raising the crosslinking capacity, molecular mass, or hydrophobicity of the polymer reduces the hydrogel’s breakdown velocity. Local cellular activity may potentially cause further breakdown of the polymeric matrix [43].

While numerous degradation methods exist, the hydrogel’s breakdown products should be biocompatible and do not cause an immune reaction. In an ideal world, hydrogel breakdown creates naturally biocompatible chemicals in the body.

## 4. Classification of Hydrogels

Hydrogels are the most commonly studied scaffolds and are categorized into origin, configuration methods of crosslinking, polymeric composition, form; Figure 3 shows the classification of hydrogels [44].

Polymers self-assemble into physically cross-linked hydrogels when subjected to changes in environmental factors including temperature, ionic concentration, etc., [45]. In contrast to chemically cross-linked hydrogels, which release heat during formation at the gelation location, physically cross-linked hydrogels cannot develop under extreme conditions such as radiation, chemical agents, or crosslinking agents. On the contrary, physically cross-linked hydrogels are less stable than chemically cross-linked hydrogels shown in Table 2.

## 5. Chemical Structure of Polymeric Hydrogels

### 5.1. Natural Polymers

Natural polymers are shown in Table 3 according to their chemical structures. 

#### 5.1.1. Polysaccharides

Polysaccharides (shown in Table 4) have been regarded as a weak cousin of the control of cellular and tissue functions in comparison to peptides and even lipids [49].

##### Chitosan

Chitosan originates from chitin by de acetylation. Chitosan is among the most extensively utilized polysaccharides in tissue regeneration, due to its outstanding bioactivity, process ability, biocompatibility, and controlled disintegration by enzymes. Chitosan is soluble exclusively in acidic solutions, and as the pH of the solution shifts from acidic to neutral, it exhibits a sol gel transition. Chitosan contains bio adhesive properties and can temporarily open a strong linkage between epithelial cells [50]. Chitosan has strong antimicrobial action against various pathogens. Xu et al. [51] fabricated an injectable and thermo-responsive hydrogel composed of chitosan/β sodium glycerophosphate (β-GP)/ gelatin that demonstrated anti-inflammatory, bone regenerative capacity and promoted periodontal regeneration. To repair intrabony periodontal abnormalities, Yan et al. [52] recommended making and testing an enzymatically hardened chitosan hydrogel in vivo. These hydrogels have a strong biodegradability and biocompatibility.

##### Alginate

Alginate is composed of repeated brown algae with a broad dispersion in the cell walls (1, 4). Alginate content, sequencing, and molecular mass vary depending on the origin. Ionic crosslinking allows alginate to form hydrogels. Alginate is biocompatible, and non-immunogenic, nevertheless, it also possesses several disadvantages, involving low cell attachment, poor mechanical strength, and low degradation rate. As a result, alginate must be modified to get the desired characteristics. For example, incorporating RGD or laminin into alginate improved cell adherence, viability, and multiplication [53]. Diniz et al. [54] created an injectable oxidized alginate microbeads enclosed with periodontal ligament, and gingival mesenchymal stem cells. For the objective of endodontics regeneration, Devillard et al. [55] created collagen/alginate-based stem cell composite frameworks. Peri-implantitis is one of the main problems with inflammation in dental implantology. RGD-alginate hydrogel incorporating stem cells was designed with good antibacterial capabilities against Aggregatibacter actinomycetemcomitans to achieve this goal [54]. In order to promote reparative procedures, Dobie et al. [56] studied the use of alginate hydrogels to deliver TGF 1 to dentin pulp complex. Human tissue cultures were treated with alginate hydrogels to start upregulating dentin matrix secretion. On cut pulpal surfaces, the hydrogels also induced odontoblast-like cell development. Alginate hydrogels served as an appropriate template to assist dental regeneration and to distribute growth factors to boost the dental pulp’s inherent capacity for regeneration.

##### Hyaluronic Acid

Hyaluronic acid is a polysaccharide naturally occurring in the body. Hyaluronic acid has an apparent viscosity, allowing it to be applied by an injection and creates a gel in the body [57]. The enzyme hyaluronidase can cleave hyaluronic acid to produce smaller molecular weight molecules. For DPSC growth, 2-aminoethyl methacrylate (AEMA)-modified hyaluronic acid (HA) (AEMA-HA) hydrogels simulate the in vivo 3D environment. Therefore, following further optimization in the future, they seem to offer enormous promise for clinical use in regenerative medicine and other therapeutic therapies [58].

##### Cellulose

Generally, cellulose is a linear polysaccharide molecule. Cellulose is a low-cost, renewable resource. It is the most plentiful natural polysaccharide. Cellulose is one of the least harmful materials on the earth, with advantages such as bioactivity, degradability, recyclability, dimensional stability, and nontoxicity [59]. Cellulose is an excellent starting material, but its usefulness is restricted, because of its problems in dissolving. Thanks to its interesting biological activities, cellulose and its variants are commonly used in biological applications [60]. In one work, B. Li et al. placed simvastatin into TiO_2_ nanotubes and subsequently overlaid a thermo-responsive chitosan-glycerin-hydroxypropyl methylcellulose hydrogel (CGHH) on top of these nanotubes. These constructions demonstrated improved osteogenesis capability at normal body temperature and antimicrobial characteristics in the presence of infection, making them viable substances for implementation [61].

##### Starch

Starch is a natural and recyclable polysaccharide that has the following characteristics: safety, nontoxicity, widespread abundance, low price, cytocompatibility, degradability, and biosafety [62]. Native starch components are including amylose and amylopectin in varying quantities (15–30 and 70–85%) depending on species, growth conditions, and collecting processes. The biochemical nature and physical arrangement of starch’s components have a serious influence on the material’s vast variety of applications [63,64,65]. By introducing extra subunits to its networks, starch may be given new functionalities and its utility can be expanded [66]. Owing to their architecture, enhanced starches have been used to carry diverse nanoparticles with therapeutic treatments against ailments in the past few decades [67]. Because of their strong hydrophilicity and viscosity stability, nonionic starch derivatives from hydroxyalkyl starch have excellent application performance. 3D-printed hydroxyapatite—Zn^2+^ functionalized starch composite has enhanced mechanical and biological features, and could be used as a bone filler in low load-bearing defects such as craniomaxillofacial bone [68].

##### Xyloglucan

For its bio adhesive and local gelation, xyloglucan has been regarded as an attractive and helpful biopolymer in drug delivery studies [69]. Xyloglucan has been widely employed in the pharmaceutical business, such as drug delivery systems. A series of publications on xyloglucan-based drug carriers for biomedical purposes, including thermo-sensitive hydrogel and thermos-reversible hydrogel in situ, was published recently [70]. As a result, this polysaccharide has several uses and should be researched further by academia. Injectable xyloglucan hydrogels could be effectively used for bone and cartilage regeneration [71]. Methylcellulose/xyloglucan mucoadhesive hydrogel exhibited a suitable syringeability to ease of administration at 25 °C. Metronidazole (MTZ) showed a more sustained release from this hydrogel than from a commercial preparation. Consequently, the in situ distribution of MTZ for the treatment of periodontitis may be accomplished using methylcellulose/xyloglucan hydrogel [72].

##### Cyclodextrin

Cyclodextrins (CDs) are cyclic oligosaccharides identified in 1903 by Franz Schardinger utilizing a bacterial decomposition of starch [73]. Because of their superior biocompatibility and absence of bio toxicity, they were employed in dental tissue regeneration [74].

##### Dextran

Dextran is a polysaccharide produced by bacterial strains that is widely accessible.

Because of its degradability, bio-compatibility, no immunogenicity, and no antigenic properties, dextran has been frequently utilized in medicine and pharmacy [75]. Glycidyl methacrylated dextran (Dex-GMA)/gelatin scaffolds containing microspheres loaded with bone morphogenetic proteins expressed outstanding alkaline phosphatase (ALP) expression, osteocalcin content, and calcium deposition. In addition, histological examination in dogs showed neo osteogenesis, and periodontal ligament regeneration. Thus, this scaffold could be used for periodontal regeneration [76].

##### Carrageenan

Carrageenans are sulfated polysaccharides with a large molecular mass isolated from red seaweeds, similar to ECM-derived glycosaminoglycans. Carrageenan was proven to have exceptionally low cytotoxicity and no mutagenicity multiple investigations [77]. Nevertheless, when synthesized in nano complexes, certain studies have raised safety concerns. A research discovered that carrageenan/chitosan NPs at proportions lying between 0.1 and 3 mg/mL had no impact on in vitro L929 fibroblasts [78]. Carrageenan’s three-dimensional architecture promotes osteoblast growth and adherence [79]. When grafted with nano hydroxyapatite, it improves osteoblast activity [80]. The addition of carrageenan to the scaffolding increases the compressive strength of the hydroxyapatite-collagen hybrid hydrogel [81]. Hybrid carrageenan hydrogels with varied nano hydroxyapatite proportions showed little cytotoxicity against human osteoblast cells and powerful antimicrobial activity versus Pseudomonas aeruginosa. Moreover, it has been demonstrated that treating cells to carrageenan nano gel and whitlockite NPs up-regulated Runt-related transcription factor-2 and OPN protein expression [82]. For the treatment of periodontitis, κ-carrageenan oligosaccharide composite hydrogels have been effectively created and shown to have antibacterial and anti-inflammatory effects. As a result, it becomes a great candidate for the treatment of periodontitis [83].

##### Gum

Natural gum polysaccharides have outstanding biological features, which have resulted in their popularity in numerous purposes. They are also more effective than synthetic and semi-synthetic materials [84]. The use of gums in the pharmaceutical industry and tissue engineering has skyrocketed according to studies. It is a popular scaffolding material because of its optimal bio-activity, degradability, and water solubility [85]. Natural gums have impressive features that enable them to be employed in nano composites, such as NPs and synthetic polymers, for cellular multiplication [86]. Human tooth-derived stem cells’ in vitro vacuolation is stimulated by gellan gum hydrogels cross-linked with carbodiimide [87]. Natural polysaccharides as scaffolds for Dental and craniofacial tissues regeneration are shown in Table 5.

#### 5.1.2. Proteins

##### Albumin

Albumin is an intrinsic, non-glycosylated protein that is primarily synthesized in the liver by hepatocellular tissues and released into the bloodstream as a substantial component of plasma. It is now commonly employed in bioengineering investigations. As a consequence, numerous studies have been carried out to explore its structure and therapeutic properties [101].

Albumin’s antimicrobial effects are being studied extensively. Albumin acts as an anti-attachment peptide, preventing biofilm formation by increasing the net negative charge and hydrophilicity of various substances [102]. Albumin-coated bone-related scaffolds were studied, and despite the fact that albumin lacks a recognized cell binding domain, it favorably promotes cellular adherence, and multiplication, hence, albumin should be included into the scaffolds to increase bone regeneration [103]. It reacts to environmental conditions, is soluble at large doses, and gelatin occurs in certain conditions.

Albumin is an appealing biomaterial for biomedical research and treatments due to its flexibility, low cost, and bioactivity. Until now, most albumin therapeutic research has focused on its usage as a carrier protein or nano particulate to optimize medication bioavailability and distribution to target areas [104]. Chitosan coating was placed on bovine serum albumin (BSA) microparticles preloaded with NEL-like molecule 1 (NELL-1) to stabilize the BSA microsphere; hence, the nanoparticle may gently liberate the NELL-1 while maintaining its biocompatibility [105]. This makes it an excellent choice for osseous tissue regeneration. Recombinant human serum albumin (rHSA) was developed to carry sodium benzoate, salicylic acid, and warfarin. The release of the first two drugs was completed in 2 h, while warfarin release was pursued for 24 h. The results showed the sequence in which medicines were able to attach to intact albumin, showing that HSA’s drug-binding capacity persisted after the hydrogel formed. Yet, fibroblast cells successfully adhered and multiplied on the hydrogel, showing that rHSA denaturation had advanced to the point where cell attachment was possible. The current rHSA hydrogel may work well as a sustained release carrier for medications with albumin affinities [106]. It is noteworthy that Li et al. [107] found that freeze-drying an albumin gel that had been crosslinked with transglutaminase resulted in a scaffold with characteristics resembling those of collagen scaffolds in terms of its physical and mechanical makeup. More importantly, the ability of the scientists to successfully differentiate human mesenchymal stem cells (MSCs) implanted in these scaffolds into osteoblasts revealed the promise of this kind of scaffolding for bone tissue creation and regenerative therapies.

##### Collagen

Collagen is one of the most often utilized bioactive scaffold. Collagen is the major organic element of numerous connective tissues, such as bone and PDL. Collagen contains motifs (e.g., Arg-Gly-Asp) that can be recognized by plasma membrane receptors to induce attachment of cells and consequent biological responsiveness; hence, it has great biocompatibility [108]. In the ECM, collagen can also serve as a growth factor repository distribution. Since self-assembled collagen hydrogels have low biomechanical toughness, crosslinking is frequently used to increase their mechanical properties. Using cross linkers (e.g., glutaraldehyde) boosts mechanical strength substantially. Cross linkers that are not biodegradable, on the other hand, may have an impact on the hydrogel’s degradation behavior and biological control. One solution is to include degradable elements, which give a more desired means to change its degradability.

Collagen gels were originally used as a supply of dermal substitute before being widely used in tissue regeneration [109]. When collagen membrane with biphasic calcium phosphate (BCP) was compared with hydroxypropylmethyl cellulose (HPMC), hydrogel membrane with BCP HPMC with BCP exhibited optimal results involving the suppression of soft tissue invasion into periodontal defect in addition to remarkable bone regeneration after 12 weeks in a canine model [110].Proanthocyanidins (PAC)-treated collagen gel, according to Choi et al., displayed increased surface roughness and improved PDL cell adhesion [111]. When used to correct class II furcation deficiencies in a canine model, collagen hydrogel scaffold injected with FGF-2 demonstrated cementum-like tissue and PDL-like Sharpey’s fiber production without ankyloses and root resorption [112].

##### Gelatin

Gelatin is a protein that has been partially hydrolyzed to produce a denatured protein. The percentage of collagen transformation into gelatin is affected by the harvesting procedure’s pH, temperature, and duration. Furthermore, because gelatin is a denatured biopolymer, using it as a scaffolding material might avoid the immunological and microbial transmission issues related to collagen. In vitro and in vivo, gelatin is biocompatible and biodegradable. Furthermore, because gelatin is a collagen derivative, it contains cell binding characteristics, which are essential for efficient biodegradation and cell attachment. It exhibits a sol-gel transformation around 37 °C, which is near to the physiologic thermal conditions. Due to this property, chemical cross linkers are used in order to enhance its architecture to be stabilized during implantation as an injectable biomaterial in vivo. Chemical treatment of gelatin has been actively investigated in contemporary times to enhance its characteristics.

Methacrylamide-functionalized gelatin (GelMA) is one option, as it preserves the cellular interaction region. In an effort to mimic the haversian canal in natural bone tissues, Zuo et al. also constructed multiple-compartment, osteon-like structures with interconnecting channels using GelMA hydrogels. Human osteosarcoma cells (MG63) and HUVEC cells grown in various construct compartments both expressed osteogenic and vasculogenic genes, according to in vitro experiments [113]. In order to rebuild bone tissues, Heo et al. reported using a hybrid material based on GelMA and GNPs [78]. Hydrogels derived from GelMA, particularly hybrid hydrogels, have been investigated as potential materials for cartilage tissue engineering. For instance, Klein’s team investigated the healing of cartilage tissues using GelMA-HA scaffolds [114,115,116]. Incorporation of IL-4 into TG gels may enhance the M2 polarization of Mφs and that SDF-1α can be employed to direct endogenous cell homing. In general, building capability for Mφ modulation and cell recruitment in high-stiffness hydrogels is a straightforward and efficient method that can enable significant amounts of periodontal tissue regeneration [117].

##### Fibrinogen

Fibrin gel is obtained by the reaction between commercially purified allogenic fibrinogen and thrombin which are the primary proteins in blood clotting [118]. According to Flanagan et al.’s research [119], the use of fibrin as a cell transport material results in autologous and mechanically durable constructs that undergo notable tissue growth in vivo. In order to create implantable constructions that may be employed in dental engineering or dentistry, fibrin may also serve as an autologous scaffolding biomaterial. A study was carried out by Kretlow et al. [120] to identify potential biomaterials that may be administered to enable the regeneration of complex tissues. Dental tissue engineering can make use of injectable biomaterials, particularly those made from aqueous solutions, which are ideal transporters for cells and other bioactive ingredients. According to Rajangam and An [121], fibrin (Fbn) and fibrinogen (Fbg) can be utilized as scaffolds because they have a tendency to achieve uniform cell distribution, a higher cell seeding effectiveness, and consistent cell distribution. They also have the capacity to migrate, proliferate, and differentiate into specific organs and tissues by releasing ECM (extracellular matrix) to produce tissues.

As a result, fibrin is the perfect substance to use as a b platform for dental tissue engineering purposes. In recognition of their unique nano and micro features, fibrin (Fbn) and fibrinogen (Fbg) have been created into various scaffolds. Various scaffolds such as hydrogels [122], nanofibers [123], nanoparticles [124], microfibers [125], microtubes [126], and microspheres [127], have been developed for a variety of tissue engineering applications, particularly dental tissue engineering.

##### Silk Fibroin

Silk fibroin (SF) is an insoluble peptide with large hydrophobic motifs released by silkworms, arachnid, and other insects that may be readily isolated and used to make sericin-free silk-derived scaffolds [128]. Because of its extremely adjustable material characteristics, great biocompatibility, and moderate foreign body reaction in vivo, silk fibroin has been employed as an outstanding framework.

Thanks to its outstanding mechanical qualities, superior biocompatibility, predictable biodegradability, and low antigenicity, SF has been extensively researched in biological and pharmacological disciplines during the last few decades. Bombyx mori silk fibroin (BSF) has recently seen widespread use as a tissue-engineered framework for the formation of bones [129]. For instance, Wang et al. [130], following a week of in vitro development, enclosed human mesenchymal stem cells (HMSC) in sonication-triggered SF hydrogels and documented multiplication and survival in stable cultures in Table 6.

### 5.2. Synthetic Polymers

#### 5.2.1. Polyethylene glycol (PEG)

PEG is a biomaterial that has been applied in many biological and pharmaceutical applications. PEG has numerous notable properties, including excellent bioactivity, biocompatibility, and anti-fouling properties, as well as solubility in water and solubility in chemical reagents [136].

Because of its high hydrophilicity, PEG is easily converted into a hydrogel via crosslinking. Chemically cross-linked PEG hydrogel has a more integrated architecture and better mechanical strength. The anti-fouling feature of PEG (also referred as the “stealth characteristic”) has been exploited to prevent the molecular and bacterial adhesion on the PEG interface; however, this feature limits cell-PEG interaction, which lowers cell adhesion to the PEG hydrogel.

To overcome this issue, bioactive substances have been included into the PEG in order to increase its cellular attraction [137]. Grafting of the PEG with natural biomolecules such as collagen can further boost its bioactivity [138]. Moreover, copolymerization of PEG with poly(e-caprolactone) (PCL) furnishes the resulting scaffold with a thermo-responsive function and improved bioactivity, making it ideal for local tissue engineering [139]. With less complicated clinical management, PEG hydrogel membrane proved equally effective in treating bone dehiscence defects surrounding dental implants as a traditional collagen membrane [140]. DPSCs can be supported by PEG-fibrinogen (PF) hydrogels, which also have the potential to influence them in ways that would be advantageous for uses in regenerative endodontics [141].

#### 5.2.2. Poly(a-Hydroxy Esters)

Poly(a-hydroxy esters), which comprise poly(glycolic acid) (PGA), poly(lactic acid) (PLA), and their copolymers PLGA and PCL, are the most often utilized artificial scaffolds in tissue engineering, since they are well defined and FDA authorized for specific therapeutic purposes. The crystallization degree of PGA is quite high. As a result, it has optimal mechanical properties, and is insoluble in most chemical reagents. PLA consists of two isomers: D-lactide and L-lactide. L-lactide is natural, while the D, L-lactide is a synthetic combination of D-lactide and L-lactide. PLLA has a semi-crystalline architecture, which results in poor solubility and great physical strength. PDLLA is amorphous and has poor mechanical properties. The copolymer of PGA and PLA is PLGA, and the GA/LA ratio affects the copolymer’s hydrophilicity and breakdown rate. Human dental pulp stem cells were biocompatible with the gelatin/PLGA-PEG-PLGA composite hydrogel. The gelatin/PLGA-PEG-PLGA composite hydrogels displayed articular cartilage-like modulus and favorable biocompatibility with chondrocytes [142]. Amoxicillin-loaded PLGA layer over titanium implant may be effectively employed to create a useful coating that has antibacterial qualities [143].

PCL is biocompatible and has solubility in a broad variety of organic media. For bone tissue engineering uses, 3D-printed PCL/GelMA hybrid scaffolds produced extremely high cell survival, osteogenic differentiation, and mineralization that were equivalent to cell culture without sacrificing mechanical strength. So with cell loading they offer a lot of potential for bone tissue engineering purposes [144].

Nevertheless, its biodegradation is slower than PGA, PLA, and PLGA. Hydrolysis degrades poly (a-hydroxy esters), and the breakdown products are typically harmless. Poly (a-hydroxy esters) do not dissolve in water and so cannot create hydrogels. However, an injectable hydrogel may be created by modifying a poly (a-hydroxy ester) with additional hydrophilic components. A PEG-PLGA-PEG triblock blend of a certain structure, for example, is a sol at ambient thermal conditions; however, it transforms into a translucent gel at body temperature [145]. Poly (a-hydroxy esters) may also be formed into injectable microspheres as cell transporters and medication delivery vehicles for tissue regeneration, in addition to hydrogels.

#### 5.2.3. Poly (N-Isopropyl Acrylamide)

Poly (N-isopropyl acrylamide) (PNIPAM) is a thermo-sensitive polymer utilized extensively in biomedical engineering. Thermosensitive injectable hydrogel containing poly (*N*-isopropylacrylamide) (PNIPAAm)-based copolymer/graphene oxide (GO) composite was developed by P. Ghandforoushan, et al. [142]. The hydrogel could promote the deposition of alkaline phosphatase (ALP) activity, in addition to increased expression of Runt-related transcription factor 2 and osteocalcin in the hDPSCs. According to this research, it is suggested that the created hydrogel be used in bone tissue engineering since it showed osteogenic capability [142].

#### 5.2.4. Pluronic Block Copolymers

Pluronics1 are poly (ethylene oxide) and poly(propylene oxide) triblock copolymers (PPO). In aqueous solution, Pluronics1 display a unique thermo-sensitive sol-gel transformation. Because of the high PEO concentration, the system dissolves in solution at low temperatures. The sol-gel transition is influenced by several characteristics, such as the PEO/PPO proportion, atomic mass, block size, and block sequencing [146]. The PEO-PPO-PEO hydrogel has structural integrity and simple to work with. Pluronic block copolymers have significant limitations as injectable biomaterials, including non-process ability and fast disintegration [147]. Copolymerization with natural material can enhance biodegradability and bioactivity [146] in Table 7.

### 5.3. Hybrid Polymeric Hydrogels

Due to the limited structural rigidity of single component hydrogels, recent investigations have employed composite or hybrid hydrogel matrices to boost hydrogel stiffness [153]. Physically cross-linked polymers such as polyurethanes and polyesters are now biocompatible, as chemically cross-linked polymers such as poly(glycerol sebacate) (PGS) and poly(citrate diol) (PCD) [154]. These degradable composites demonstrated highly adjustable degradation, mild biocompatibility, and excellent mechanical performance [154]. Because of their limited structural rigidity or weak biocompatibility, they have shown intriguing uses in tissue regeneration [70]. Designing hybrid polymers to generate biodegradable elastomers with optimal qualities to fulfil distinct tissue-specific needs has become an appealing alternative for making these elastomers successful for a larger range of biomedical applications.

PGS-PCL hybrid elastomers have been successfully created via solvent electro spinning. The addition of PCL considerably improved the creation of the nano fibrous structure, and the hybrid materials demonstrated mechanical capabilities comparable to human aortic valve tissues [155]. In order to create hybrid polymers for tissue regeneration, gelatin was additionally mixed into PGS elastomer.

The inclusion of gelatin improved the mechanical characteristics and bioactivity of PGS elastomers substantially [156]. Osaheni et al. [157] designed poly([2-(methacryloyloxy)ethyl] dimethyl-(3-sulfopropyl) ammonium hydroxide) with varied quantities of biocompatible zwitterionic polymer film, indicating that biocompatible zwitterionic polymers decreased friction by up to 80%. This scaffold can be used in dentistry to reduce friction and wear on dental materials [158]. Different types of applications of hydrogel materials are shown in Table 8.

## 6. Composition of Polymeric Frameworks

Some major kinds of hydrogels are formed as a result of the synthesis process.

The following are common examples: homopolymeric hydrogels, copolymeric hydrogels, and multipolymers etc.

Homopolymeric hydrogels are composed of a single component [172]. Based on the monomer and synthesis technique, homopolymers could have a cross-linked structural framework.

Copolymeric hydrogels are formed of two or more different types [173].

Multipolymers are fascinating groups of hydrogels that are composed of two separate cross-linked synthetic and/or natural polymer components entrapped in a matrix. One portion of the hydrogel is a cross-linked polymer, while the other ingredient is a non-cross-linked polymer [174].

## 7. Configuration of Polymeric Matrices

Hydrogels can be classified based on their morphology and structure as following [175]:(a)Amorphous.(b)Semicrystalline.(c)Crystalline.

## 8. Gelation Methods

There are many strategies for crosslinking of polymers.

### 8.1. Physically Crosslinked Hydrogels

#### 8.1.1. Ionic Crosslinked Hydrogels

Ionic crosslinking is a type of physical crosslinking that involves combining ionizable polymers with di- and/or trivalent cations. Other cations that can be employed as crosslinkers include Sr^2+^, Ba^2+,^ and Zn^2+.^

The ion selection affects more than only permeability and crosslinking [176,177], but also the emission of the polymer’s composition [178]. To create ionically crosslinked hydrogels, a dual syringe applicator is employed. Whenever the polymeric solution comes into contact with the cation solution, and gelation occurs. Alginate is an ionizable polysaccharide. The gel formation rates, and mechanics of the alginate hydrogel are influenced by the M/G block ratio, in addition to the ionic strength of the solution and the temperature [179]. Calcium chloride is the most implemented crosslinker for alginate, and it crosslinks carboxylic groups in the alginate to produce a “egg box”-like shape [180]. Raising the density of crosslinking by adding significant quantities of divalent cations may conflict with numerous biological processes, affecting tissue development [181]. Alginate’s gradual crosslinking reaction improves its mechanical integrity and structural uniformity.

One disadvantage of employing ionically crosslinked alginate is its short-term stability in physiological settings. Calcium-crosslinked alginate has been demonstrated to have considerable fluctuations in degradability, and an unpredictable in vivo degradability is a key drawback of employing alginate hydrogel [182]. Furthermore, because alginate lacks unique cellular activity, the inclusion of bioactive molecules (e.g., RGD) is required for the appropriate biological functions. Pectin, like alginate, is a biopolymer with a carboxylic group that may be crosslinked by cations [183]. Pectin hydrogels, like alginates, have in adequate resorption and can disintegrate under physiological circumstances. Poly[di(carboxylatophenoxy)phosphazene] is another ionically crosslinked hydrogel [184]. Ionic crosslinked hydrogels are also generated by the ionic interaction of cationic and anionic polymers.

#### 8.1.2. Hydrogen Bond Crosslinked Hydrogels

Hydrogen crosslinking has been discovered in several bioactive molecules; for instance, hydrogen bonding holds the double strands of DNA together [185]. Hydrogen bonding occurs in a hydrogel when electron-deficient hydrogen contacts with an area of high electron density [186]. Several parameters impact the gelation process, including temperature, polymer proportion and concentration, nature of the reagent, and proportion of interaction between polymer functions [187]. A hydrogen bond-crosslinked hydrogel cannot attain strong structural integrity on its own, and a chemical crosslinking technique is frequently required. A double hydrogen bonding hydrogel was recently reported to have high tensile and compressive strengths over a wide pH range [188]. Moreover, a unique manufacturing approach for producing a robust hydrogen bond, crosslinked hydrogel without the need of chemical inducers or covalent bonding crosslinking agents was devised by copolymerizing poly(N-vinylpyrrolidone) and acrylamide [189].

#### 8.1.3. Thermally Triggered Hydrogels

Temperature-induced crosslinking hydrogels are biomaterials that self-gelate in response to temperature changes. Temperature-induced crosslinked hydrogels are classified into two types based on their response to changes in critical temperature. When the temperature rises over a critical point, one set of polymers forms gels, while the other group dissolves. These two groups’ critical temperatures are referred to as “lower critical solution temperatures” (LCST) and “upper critical solution temperatures (UCST),” respectively. Gelatin and some polysaccharides, such as carrageenan, have UCST properties. The creation of double (e.g., polysaccharides) or triple helices drives the formation of helical aggregations (e.g., gelatin) [190,191]. Copolymers comprising PNIPAM, Pluronics1, PEO/PLGA, and PEG-based amphiphilic block copolymers are examples of LCST biomaterials [192]. The LCST polymer is useful for in situ gelation. Polymer solutions should be fluid at ambient temperature and gel at biological temperature. Thermally triggered crosslinked hydrogels have various advantages, including the absence of the requirement for organic crosslinkers and initiators, as well as no thermal influence on neighboring tissues. Temperature-induced crosslinking hydrogels, such as other physically crosslinked hydrogels, sometimes require chemical crosslinking to improve injectable scaffold durability.

### 8.2. Chemically Crosslinked Hydrogels

Chemically crosslinked hydrogels are made up of covalent connections between polymer chains. Chain growth propagation, addition, and condensing polymerization, and gamma and e–beam gelation are all used to create polymerized hydrogels. In most circumstances, the crosslinking procedure produces an irreversible hydrogel with greater biomechanical features than a physically crosslinked hydrogel. Nevertheless, covalent connections in certain hydrogel systems may be destroyed and reformed in a reversible way. These hydrogels are known as reversible chemically crosslinked hydrogels, and they have longevity as well as local flexibility [193,194].

## 9. Classification of Hydrogels as Various Hydrogel Structure Used in Bone Regeneration

The development of a feasible hydrogel composition capable of encapsulating and delivering peptides and bioactive compounds is required for hydrogel-based bone healing.

Various manufacturing processes may be used to create a wide range of hydrogel structures. Continuous refinement of manufacturing techniques to generate acceptable hydrogel formulas for healing bone defects is required to increase the implementation of hydrogels in tissue regeneration. Knowledge of intricate methods of hydrogel manufacturing and changing the framework to improve biocompatibility and the hydrogel’s osteoconductive, osteoinductive, and osteogenic characteristics will speed up the development of viable configurations. We will go over the development, production, benefits, and drawbacks of three different hydrogel frameworks: microbeads, nanoparticles, and hydrogel fibers.

### 9.1. Hydrogel Microbeads

Microbeads of polymeric hydrogels are utilized for bone tissue regeneration and can be created by microfluidics, emulsification, electrostatic droplet extrusion, coaxial air jetting, and in situ polymerization.

Traditional procedures, on the other hand, do not produce homogeneous small-sized microbeads.

Researchers have devised a non-equilibrium microfluidic process for the fabrication of smaller-sized hydrogel beads in recent times (size less than 100 mm).

Polymer materials are implanted into a nonequilibrium W/O interaction comprising hydrogel substances, where the water molecules are dissolved into a continuous phase and hydrogel precursors in water-in-oil particles quickly got smaller and condensed, forming microbeads that are smaller than those created by the standard approach [195]. Furthermore, the morphology of hydrogel microbeads may be adjusted by changing the degree of droplet shrinkage and the quantity of crosslinker.

Smaller-sized microbeads offer an advantage over larger-sized microbeads in achieving optimum cell assembly.

Because of the large elevation in surface volume fraction in relation to standard hydrogel, the relatively small hydrogel microbeads have higher mass transmission rate, which is advantageous for medication and stem cell administration to bone defect locations [196]. Moshaverinia and colleagues created an injectable alginate hydrogel microbead to contain dental-derived MSCs such as gingival mesenchymal stem cells (GMSCs) and periodontal ligament stem cells (PDLSCs) [197]. The alginate and stem cell combination was syringe-squeezed into a calcium chloride solution for ionic crosslinking.

Micro-CT examination revealed that the cells stayed alive after transplantation, and aberrant calcification was found both within and outside the microbeads as a result of effective nutrition and oxygen delivery.

However, because the in vivo nonenzymatic breakdown of alginate microbeads takes so long, Leslie and colleagues [198], it was planned to include alginate lyase into microbeads to manage the alginate breakdown.

The microbeads were made with a high electrostatic potential without affecting the osteogenic differentiation capability of the encapsulated rat adipose-derived stem cells (ASCs).

When compared to microbeads that did not contain ASCs or were not subjected to alginate lyase, elevated concentrations of osteocalcin expression were detected.

Wang et al. utilized chitosan and collagen to fabricate hydrogel microbeads via double crosslinking mechanisms, which exhibited upregulation of signaling pathway osterix and osteocalcin and considerable accumulation of bone mineral deposits within the osteogenic medium [199]. Wise and colleagues conducted another intriguing study in which they used a water-in-oil emulsion method to create collagen-chitosan hydrogel microbeads to encapsulate MSCs and BMMCs, which showed a synergic activity in boosting mineral accumulation and improving ectopic bone regeneration [200]. The findings demonstrated that microbeads comprising both MSCs and BMMCs in osteogenic medium (group F C O) had impressive bone volume.

### 9.2. Hydrogel Nanoparticles

Hydrogel nanoparticles (nanogels) are spherical nanoparticles created by physically or chemically cross-linking three-dimensional polymeric matrix that may expand in water.

Nanogels are often produced using emulsion polymerization methods such as reverse emulsion polymerization and distillation-precipitation polymerization, which copolymerizes a quickly agitated liquid at high temperatures to produce stable suspensions [201]. Nanogels exhibit a number of hydrogel qualities, such as strong biomechanics, and are extremely useful in bone regeneration applications. Nanogels have the texture, dimension, and other characteristics of nanoparticles. Because of their homogeneity, customizable size, simplicity of design and manufacturing, vast surface area of multivalent biologic conjugation, high drug loading capability, and superior entrapment integrity, they are potential responsive vehicles for targeted drug administration.

Nanogels can also transport peptides without conformational changes using the original encapsulating approach under mild processing conditions. According to a recent work [202], hydrogel nanoparticles made of hydrophobized cholesterol-bearing pullulan (CHP) can be utilized to transport a range of hydrophobic peptides and enzymes. Through combined treatment, acrylate group modified CHP nanogels administered recombinant human fibroblast growth factor 18 (FGF-18) and recombinant human BMP-2 to bone lesions and efficiently stimulated bone cells to rebuild bone [202]. This demonstrates that the nanogels may efficiently transport two distinct proteins and stimulate bone healing.

Miyahara et al. employed collagen membrane, cholesterol-bearing pullulan (CHP) nanogel membrane, and an untreated membrane in the repair of adult Wistar rats’ parietal bone defect. Four weeks later, scientists discovered that the rat in the nanogel group had much more new bone development than the other two groups.

Seo and colleagues created nanogels with dimensions less than 200 nm that gelled promptly after local administration [203]. Nanogels made of hydrophobic isoleucine ethyl ester and hydrophilic polyethylene glycol were discovered to regulate the release of bone BMP-2 via a hydrophobic contact and an ionic interface between BMP-2 and carrier materials. While nanogels have showed efficacy in delivering proteins and growth factors and have significant potential in efficiently inducing bone formation, building adjustable hydrogels with excellent mechanical durability and prolonged release is critical in developing viable therapies for bone repair.

### 9.3. Hydrogel Fibers

Hydrogel fibers have a porous structure and a diameter ranging from a few nanometers to a few microns [204]. Hydrogel fibers have a porous structure and a diameter ranging from a few nanometers to a few microns, electrospinning [205], wet spinning [206], gel spinning [207], 3D printing technology [208], and hydrodynamic spinning [209], among which electrospinning and microfluidic spinning are currently most widely investigated. Fibers must be further crosslinked in the presence of heat or UV radiation, glutaraldehyde, or enzymes to create hydrogel fibers [210]. Hydrogel fibers, as opposed to microbeads, may be aligned axially with the syringe and injected into the defect site; moreover, they can stay at the implantation site for a prolonged length of time [211]. Because of its high surface-to-volume ratio, quick reaction, and immobilization capabilities, hydrogel fibers have showed promise in tissue engineering.

Highly porous poly(-caprolactone) fibrous mesh(es) with well-controlled 3D architecture were tested and found to have favorable cellular reactions, noticeably increased levels of mineral deposits, increased expression of osteogenic-related genes, and osteogenesis in biologically active amorphous magnesium phosphate-laden gelatin methacryloyl hydrogels [212]. Injectable alginate-fibrin fibrous hydrogel packed with calcium phosphate cement incorporating induced pluripotent, dental pulp, and bone marrow stem cells was evolved, which resulted in increased stem cellular multiplication, calcification, and osteogenic differentiation, and may improve bone regeneration in dental, maxillofacial, and orthopaedic implementations [213].

Advancement in eliminating hydrogel fiber’s inherent shortcomings, including low mechanical qualities and fast release, is critical to producing a hydrogel formula that allows guided sustained release of administered peptides and medicines.

### 9.4. Injectable Hydrogels

Injectable scaffolds offer many advantages in tissue regeneration over pre-formed scaffolds [214] and are employed for the regeneration of craniofacial and dental tissues [39,215,216]. Numerous investigations [217,218,219,220] have found that tissue engineering strategies utilizing biomaterials may replace absent or compromised craniofacial and dental structures in addition to restoration of their biological functions. There are several mechanisms that contribute to a hydrogel’s injectability. These mechanisms can be categorized into three groups: locally gelling liquids, or solutions or liquids that typically flow but stiffen into gels when injected into the body, make up the first mechanism. The second technique involves injectable gels. Although certain gels are produced ex vivo, they can be injected, owing to their shear thinning properties and ability to recover their hydrogel structure after relaxing operations. In addition to the two types of injectable systems already mentioned, the third class of injectable particles may also be injected while submerged in a liquid phase [221].

## 10. Application of Hydrogels in Dental and Osseous Regeneration

Bioactive hydrogels are widely implemented in the regeneration of both dental and osseous tissues [222], as shown in Figure 4.

### 10.1. Dentin-Pulp Complex Regeneration

The dentin-pulp complex is a remarkable tooth component that comprises both hard and soft tissues. Extensive cavities and trauma, in particular, harm the dentin-pulp system, and if left untreated, this impairment progresses to permanent pulpitis. Several approaches have been adopted to recreate the dentin-pulp complex coupled cells using frameworks such as PuraMatrix (BD Biosciences, Sigma Aldrich, New York, NY, USA), nano fibrous gelatin/silica bioactive glass hybrid, collagen, and poly-L-lactic acid [223].

Human dental pulp stem cells (hDPSCs) and human umbilical vein endothelial cells (HUVECs) were entrapped in a hydrogel containing 5% gelatin methacrylate (GelMA) and implanted into tooth root segments (RSs). GelMA enhanced hDPSC/HUVEC cell adhesion and multiplication, as well as infiltrating host cell adhesion, according to immunofluorescent (IF) histochemical examination [224]. Signaling molecules and injectable biodegradable polymer templates have sped up practical implementation and improved dental tissue engineering.

Regenerative endodontics is predicted to be the best physiological strategies in regenerative dental care depending on the size and functionality of the dentin-pulp complex [225,226].

### 10.2. Cementum Regeneration

Cementum is made up of 50% organic matrix, 90% of which is collagen 1 (COL1). Type III collagen is another cement-related collagen protein that is implicated in the formation, maintenance, and regeneration of periodontal tissues [227]. Chitosan hydrogel (CsHG) was developed and implanted into rat intrabony periodontal deformities with or without fluorescently labelled PDL cells; unmanaged defects served as controls. The findings demonstrate that PDL cells stayed functional after 4 weeks of complexation within CsHG prior to transplantation, but histological analysis revealed that the CsHG were diminished without affecting the adjacent tissue. CsHG without cell loading, on the other hand, had shown a newly established cellular cementum restricted to the apical part of the defect [228]. Chitin-poly(lactic-co-glycolic acid) (PLGA)/nanobioactive glass ceramic (nBGC)/cementum are used to make a highly permeable multi nanostructured hydrogel framework loaded with growth factors. Histopathological and immunohistochemistry examinations support the development of fresh cementum, fibrillar PDL, and alveolar bone with well-defined bony trabeculae [229].

### 10.3. Gingival Tissue Regeneration

Gingival tissue can be fully restored [180,230]. Gingival tissue is a fibromucous membrane that surrounds the mandibular and maxillary bone and is created to endure constant chewing forces. Clinically, it is classified as free, inserted, or papillary. It is composed of stratified squamous epithelium that is densely packed with keratinocytes. The gingival epithelium is separated into three layers: the oral epithelium, the sulcus epithelium, and the junction epithelium [231]. It has been utilized to rebuild the interdental papilla in the field of dental aesthetics, where it is entrapped until fluid overload, and after 3 months of follow-up, a fully healthy gingiva was revealed and the papilla was occupied [232].

### 10.4. Periodontal Regeneration

Periodontal disease is an inflammation caused by intricate interactions between the human immune response and the microorganisms found in plaque. Periodontitis-induced alveolar bone resorption is regarded as one of the leading causes of teeth loss in adults. According to one study, chitosan (CS), -sodium glycerophosphate (-GP), and gelatin were combined to create an implantable and thermo-responsive hydrogel capable of constantly releasing aspirin and erythropoietin (EPO). In vivo studies demonstrated that the hydrogels are helpful in reducing irritation and promoting periodontal regeneration. These findings suggest that our manufactured hydrogels have a high prospects for development of therapeutic applications [233]. A composite hydrogel consisted of polyvinyl pyrrolidone (PVP), polyvinyl alcohol (PVA), and polyethylene glycol (PEG) blends was loaded with 2 IU/g of insulin and demonstrated favorable regular prolonged release for nearly 13 days, making it an attractive local carrier for the consistent delivery of insulin within periodontal pockets [234].

Bone morphogenetic protein-7 (BMP-7) and ornidazole (ORN)-loaded chitosan/β- as an injectable hydrogel and glycerophosphate (CS/-GP) thermosensitive hydrogel were designed and evaluated in class III furcation abnormalities in dogs. The findings of this experiment demonstrated an increase in the number of osteoblasts, with a corresponding decline in the amount of osteoclasts, and exhibited clear antibacterial action against *P. gingivalis*. These findings show that this approach might be used in periodontal treatment [235]. In mice, an injectable and thermo-responsive chitosan/gelatin/glycerol phosphate supplemented with BMP-6/iPSCs offered a 3D structure for transplanted stem cells as well as improved stem cell transport and colony formation, reduced inflammation, and promoted periodontal regeneration [236].

### 10.5. Bone Regeneration

Traumatic or surgical bone deformities play a major part in bone disunion, prolonged union or nonunion, and localized malfunction. Autografting or allografting is now the most successful treatment for bone abnormalities, but it is restricted by a protracted treatment process, highly specialized prerequisites, postoperative problems, and a scarcity of suitable donors, to ensure optimal and efficient treatment. Because of their capacity to induce cellular recruitment and development for bone repair, biomaterial scaffolding are being evaluated as a viable therapeutic replacement [237].

Because of their porous structure, which facilitates cell penetration, angiogenesis, and osteogenesis, functionalized hydrogel scaffolding have been designed as an efficient scaffold for bone regeneration [238]. Controlling the differentiation of stem cells into osteoblasts for bone production is one technique for bone tissue regeneration.

The hydrogels are capable of loading and delivering stem cells and osteoblasts. In cranial deformities, a composite hydrogel consisting of silk nanofibers (SNF) and hydroxyapatite nanoparticles (HA), deferoxamine (DFO), and bone morphogenetic protein-2 (BMP-2) was investigated. The scaffold employed demonstrated angiogenesis and osteogenesis results that expedited the regeneration of functional bones toward normal bone conformation and structure. The findings indicate that these hydrogels may have therapeutic applications in bone tissue engineering [239].

### 10.6. Cartilage Repair

Cartilage is a kind of connective tissue that is stretchable. The primary maxillofacial cartilages include the nasal septal cartilage, ear cartilages, and TMJ condylar cartilage. Because cartilage is avascular, regeneration of fresh cartilage is difficult. Hydrogels have been studied for articular cartilage regeneration because they imitate cartilaginous matrices. There have been few studies on the regeneration of nasal and auricular cartilage tissue. The clinical demand for tissue-engineered cartilage is enormous and clinically significant. Traumatic and deteriorating articular cartilage defects are the major reasons for disability [240]. Tissue engineering technologies, such as the implementation of hydrogel frameworks, to increase cartilage repair and regeneration will consequently have a significant therapeutic potential. The benefit of injectable treatments for cartilage regeneration is that the implantation not only remains within the lesion, but also permits rapid weight-bearing owing to the strength and durability that is developed nearly quickly.

Furthermore, one benefit of injectable therapy over surgical intervention is the use of less invasive procedures. Adipose-derived stem cells can be delivered via alginate microgels [241]. Because alginate microgels degrade slowly, alginatelyase is used to facilitate a significantly quicker cell release. Since cartilage is avascular and lacks innervation, regeneration and restoration are more difficult. Supramolecular microgels are especially well-suited for preserving cell viability and increasing cell differentiation of implanted mesenchymal stem cells (MSCs), which can improve cartilage repair. MSCs can be stimulated by physically crosslinked chondroitin sulphate and chitosan embedded in agarose microbeads that imitate the extracellular matrix of cartilaginous tissue. MSCs can survive, multiply, and express chondrogenic differentiation genes in the agarose microbeads. Injuries to the articular cartilage are frequently associated with inflammatory disease and discomfort. As a result, anti-inflammation therapy has emerged as an unique approach to treating articular cartilage damage, Daley et al. [242]. Human MSC- encapsulated alginate microspheres and transforming growth factor-3 (TGF-3)-containing alginate microgels were produced for cartilage repair.

TGF-3 and human MSCs can minimize apoptosis and enhance optimal cartilage matrix synthesis after hydrogel implantation. Injectable xyloglucan biomaterials containing adipose stem cell spheroids promote cell growth and dissemination, as well as effective differentiation into osteoblasts and chondrocytes, making it a feasible approach for bone and cartilage repair [243]. The alendronate-loaded nano-hydroxyapatite/poly(l-glutamic acid)-dextran (nHA/PLGA-Dex) hydrogel was produced and injected into a rat skull lesion. The TMJ is a fibrocartilage that connects the jaw to the temporal bone.

The regeneration of the mandibular condyle must combine bone and cartilage components in an osteochondral construct that is morphologically, physiologically, and mechanically identical to the native TMJ condyle.

This type of structure can be made from multilevel or bi-layer hydrogels [244]. For example, rat bone marrow mesenchymal stem cells were stimulated in vitro to develop into chondrocytes and osteoinductive cells before being enclosed in a PEG-based a double layer hydrogel and transplanted in vivo [245]. The results revealed that the hydrogel exhibits adhesion properties and anti-inflammation properties, as well as the capacity to induce osteogenesis and shows substantial potential for bone regeneration applications. After administration into a major sized cranial defect animal, adipose stem cells (ASC) were encased in an injectable gelatin hydrogel, which improved ASC viability and expedited bone healing.

## 11. Conclusions and Future Perspectives

We detailed the treatment techniques of dental and osseous regenerative hydrogels in this study, which encouraged the regeneration of these tissues to some level. Enhancing stem cell development and differentiation is a study focus in the domain of dental and osseous regeneration. Hydrogels, by imitating the microenvironment of ECM, can create a favorable development platform for osteoblasts, chondrocytes, and fibroblasts, as well as stimulate dental and osseous regeneration. Natural polymers, synthetic bioactive constituents, and tissue regeneration compounds can be used in the creation of hydrogel scaffolds to enhance cell proliferation and differentiation. Nevertheless, mechanical characteristics are insufficient when using natural polymers as a hydrogels matrix.

To increase its mechanical characteristics, several design elements must be included, including sacrificial bonds, the development of homogenous networked hydrogels, the creation of double-network hydrogels, and the inclusion of filler particles. Angiogenesis will deliver vital signaling molecules and power to the tissue location, allowing future bone repair to proceed easily. Nevertheless, loading angiogenic elements is a significant problem, and we must work hard to assure its viability. Some negative effects induced by its high proportion in the human body must be limited. Chemokines may be a better alternative than growth factors since they are indirect control rather than direct involvement in angiogenesis, and their influence is lower.

When employing implantable biomedical materials, the allergic reaction must be taken into account. Because immune response is responsible for many cases of inadequate therapeutic efficacy, treating immune response is crucial. We may also add anti-inflammatory medications into hydrogels by drug loading to generate anti-inflammatory effects. Nevertheless, we must be mindful of medicine side effects and dose. To some degree, inflammation aids in eventual tissue regeneration. Promoting mineralization is an excellent method for dental and osseous regeneration. As a site for crystal mineralization, hydrogels should mimic the extracellular matrix. Mineralization is often induced by the addition of bioactive ceramic components.

The molecular chemistry of hydrogels must be considered because some constituents can function as calcium ion binding sites and impact the structure of crystalline particles. The features of dental and osseous regeneration hydrogel therapy approaches are summarized above.

Because of their ability to disseminate in a less intrusive manner and other advantages, polymeric hydrogels are encouraging craniofacial bone engineering. For these uses, a variety of artificial, and natural substances have been tried. When implemented for dental and craniofacial tissue regeneration, such bioscaffolds outperformed conventional biomaterials. These new biomaterials point the way forward for the next generation of injectable biomaterials and need further investigation.

A range of characteristics, such as gelation time, cytotoxicity, biocompatibility, degradation, and mechanical qualities, should be addressed when creating an injectable scaffold from a biomaterial. Each factor has an effect on cell attachment, multiplication, development, tissue deposition, and the host reaction.

The crosslinking density regulates the pore size inside the hydrogel network, which influences the survival and multiplication of the embedded cells as well as the pharmacokinetic profile within the hydrogel. The gelation procedure should take place under modest circumstances to prevent harming the integrated bioactive compounds and cells. The introduction of active units into a hydrogel’s polymer helps the crosslinking reaction; nevertheless, the functional groups may exhibit interactivity with added biological molecules during the crosslinking process.

All of these findings emphasized the need of taking a holistic approach when designing or selecting a biomaterial scaffolding. One of the most significant obstacles to effective tissue regeneration is regenerating a tissue with the right structure. Future hydrogel development methodology will not be confined to these designs, but will integrate their qualities to create multipurpose hydrogels. On the other hand, extracellular matrix multitude of environmental research should be prioritized. Upcoming tissue regeneration hydrogels will most likely govern the bone regeneration process throughout time. This will necessitate more subtle modifications to dental and osseous regenerative hydrogels. To summarize, one of the future research trends is the development of multi-function hydrogels that imitate the ECM microenvironment. However, ongoing research into single-function hydrogels should be conserved since it will provide us with more options for dealing with various instances. Finally, hydrogels show considerable promise in the therapy of bone deformities. We can provide more suitable treatments for patients with dental and osseous deficiencies by adopting absolute tissue regeneration hydrogels as research on dental and osseous tissue regeneration advances.

## Figures and Tables

**Figure 1 pharmaceuticals-16-00702-f001:**
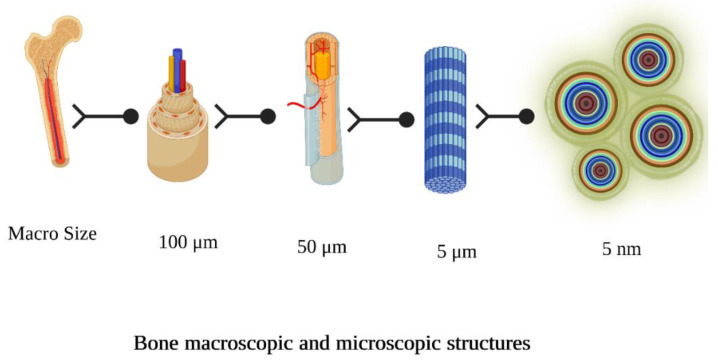
Bone macroscopic and microscopic structure.

**Figure 2 pharmaceuticals-16-00702-f002:**
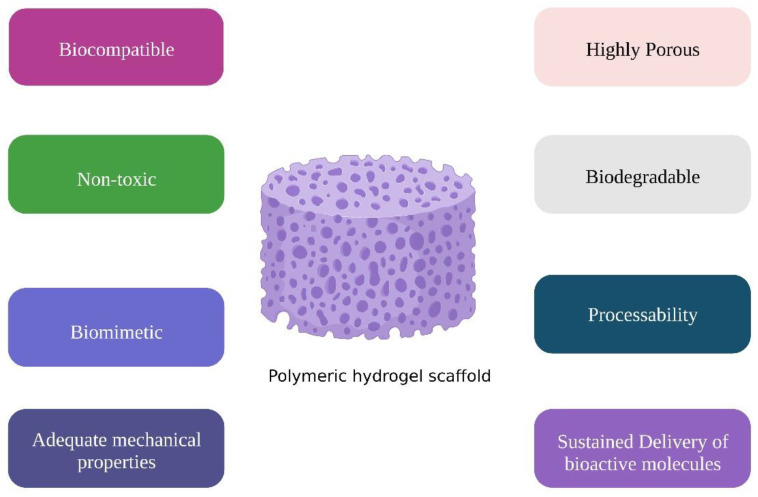
Requirement of tissue engineering scaffold.

**Figure 3 pharmaceuticals-16-00702-f003:**
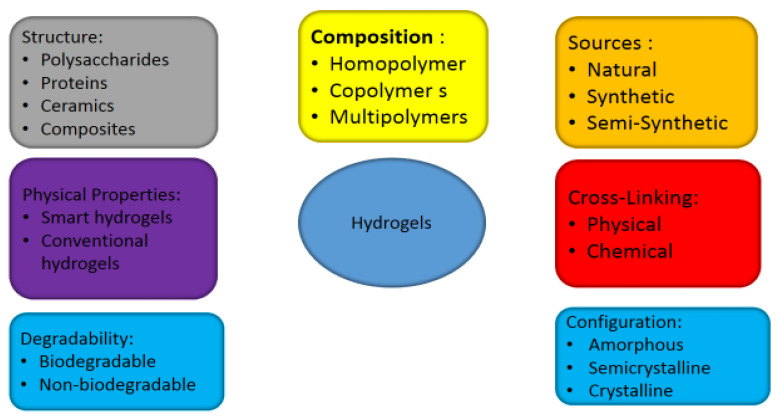
Classification of hydrogels.

**Figure 4 pharmaceuticals-16-00702-f004:**
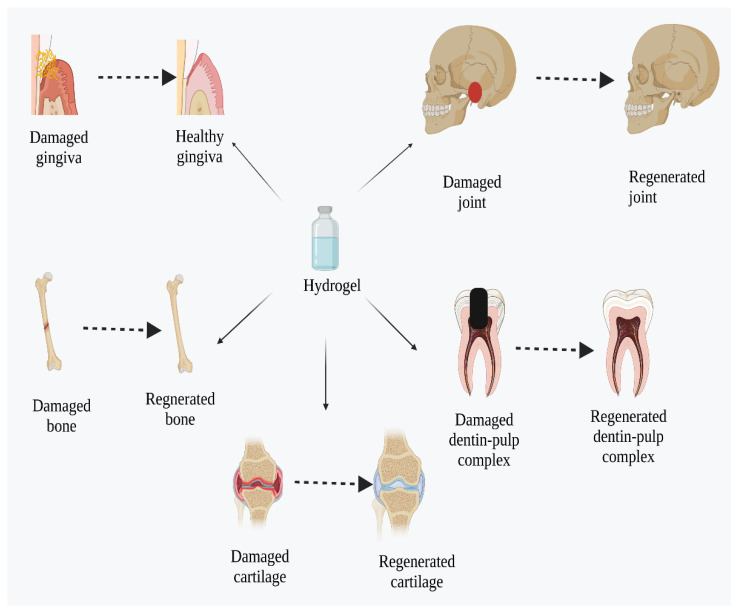
Hydrogel-assisted regeneration in dental and osseous tissues.

**Table 1 pharmaceuticals-16-00702-t001:** Classification of biomaterials implemented in tissue engineering.

Natural polymers	- Polysaccharides; e.g., chitosan, alginate, hyaluronic acid etc. - Proteins; e.g., collagen, gelatin, albumin, etc.
Synthetic polymers	Polyglycolic acid, polycaprolactone, etc.
Ceramics	Hydroxyapatite, bio glass, β-tricalcium phosphates, etc.
Hybrid	- Natural polymers with synthetic polymers; e.g., chitosan and PLGA(poly(D,L-lactide-co-glycolide)). - Natural polymers with ceramics e.g., chitosan and hydroxyapatite - Two or more artificial scaffolds e.g., PEG (polyethylene glycol) and PLGA - Synthetic polymers with bio ceramic e.g., PLGA with hydroxyapatite
Metals	Gold, silver, titanium, etc.

**Table 2 pharmaceuticals-16-00702-t002:** Advantages of polymeric hydrogel for dental and osseous regeneration [46].

Application	Advantages
Drug delivery	- Improved medication protection and stability - Long-term drug retention and controlled drug release - Responsive medication release in responding to environmental cues such as pH
Detoxification	- Detoxification agent restriction to disease site - Extended accumulation and continuous release of degradable materials
Immune modulation	Unintended consequences Elimination Therapeutic and medication doses were regulated; and cargo was released in response to internal and environmental cues.
Tissue engineering	Excellent physical and mechanical qualities Localized and regulated medication distribution Increased bioavailability

**Table 3 pharmaceuticals-16-00702-t003:** Classification of natural polymers according to their structure [47,48].

Polysaccharides	Proteins
Alginate	Collagen
Starch	Gelatin
Cellulose	Silk
Chitosan	Fibrin
Cyclodextrin	Albumin
Dextran	
Gum polysaccharides	
Pectin	
Pullulan	
Heparin	
Chondroitin sulfate	

**Table 4 pharmaceuticals-16-00702-t004:** Classification of polysaccharides according to their origin [46].

Plants	Mucilage, Pectin, Hemicellulose, Gums Cellulose, Glucomannan, Starch
Algae	carrageenans, alginates
Animals	cellulose, glycosaminoglycans, hyaluronic acid, chitosan, Chitin
Bacteria	cellulose, Dextran
Fungal	yeast glucans, chitosan, chitin, pollulan, Elsinan

**Table 5 pharmaceuticals-16-00702-t005:** Natural polysaccharides as scaffolds for Dental and craniofacial tissues regeneration.

Natural Polysaccharide	Delivery System	Outcome	Ref
Carrageenan	Nano-HA/gum Arabic/k-carrageenan composite	Without cytotoxicity, osteoblast-like cells exhibit substantial osteogenic signals.	[88] 2020
Carrageenan	Ag/carrageenan/gelatin nanocomposite	The unique Ag/carrageenan-gelatin hydrogel’s antimicrobial, drug carriers, and antitumor capabilities	[89] 2021
54646N-carboxyethyl chitosan/ hyaluronic acid-aldehyde	N-carboxyethyl chitosan/hyaluronic acid-aldehyde loaded with nano hydroxyapatite	preserving alveolar ridge integrity and facilitating soft tissue healing process	[90] 2020
regenerated cellulose (rCL) nanofibers/chitosan (CS)	Regenerated cellulose (rCL) nano fibers chitosan (CS) hydrogel	The rCL/CS scaffold aided bio mineralization and increased the survival, adherence, and multiplication of preosteoblast cells (MC3T3-E1).	[91] 2021
Chitosan/hyaluronic acid	Chitosan/hyaluronic acid nano pearl composite	RUNX2, OCN, and OPN gene expression increases. The best results were achieved with 10 wt% and 25 wt% nano pearl.	[92] 2020
chitosan	Chitosan nanohydrogel/poly-ε-caprolactone (PCL) loaded with nanotriclosan and flurbiprofen	The combination of anti-microbial and anti-inflammation properties resulted in a remarkable treatment outcome.	[93] 2019
Gelatin/Alginate	Gelatin-alginate-graphene oxide nano framework	Amplification of the transcription of osteoblast enhancing factors and ALP	[94] 2019
Carrageenan	Carrageenan/whitlockite nano composite hydrogel	Improved osteogenic development and ALP expression	[82] 2019
Carrageenan	Carrageenan/nanohydroxyapatite composite scaffold	Enhancement of osteogenic differentiation without the use of pharmacological drugs	[95] 2018
Chitosan	Chitosan gold nanoparticles mixed with peroxisome proliferator-activated ligand	Optimizing the outcome of implant placement in diabetic patients (bone development and mineralization)	[96] 2017
Alginate/chitosan	Alginate/chitosan loaded with nanohydroxylapatite	Increased hydroxyapatite levels stimulate MC3T3 cell development and calcification.	[97] 2015
Alginate	Alginate hydrogel mixed with bovine dental pulp extracellular matrix (pECM)	Accelerated differentiation in the mineralizing environment lead to mineralization at the hydrogel’s perimeter. HA hydrogels integrating PL enhanced cell functions and hDPSC mineralized matrix formation.	[98]
Hyaluronic acid hydrogel	Photo crosslinking of methacrylated HA incorporated with PL	Accelerated differentiation in the mineralizing environment lead to time-dependent mineral deposition at the hydrogel’s perimeter. HA hydrogels integrating PL enhanced cell functions and hDPSC mineralized matrix formation.	[99]
Chitosan	Ag-blended bioactive glass micro particles mixed with chitosan (Ag-BG/CS).	Ag-BG/CS enhanced the odontogenic differentiation capability of lipopolysaccharide-increased inflammatory reactivity in dental pulp cells and shown antimicrobial and anti-inflammatory activities	[100]

**Table 6 pharmaceuticals-16-00702-t006:** Natural protein in sonication-triggered SF hydrogels.

Natural Protein	Delivery System	Experiment Design	Outcome	Reference
Collagen	Collagen hydrogel loaded with Rat pulp cells marked with indium-111-oxine	Implantation in the rat emptied pulp chamber.	Functioning fibroblasts, neovasculature, and nerve fibers were seen in the collagen hydrogel one month after insertion.	[131]
Collagen	Blends with nano keratin, and hydroxyapatite	Histomorphometry on critical size defects in rat calvaria	bio-compatibility, biodegradability, and increased density of newly formed bone	[132]
Gelatin	Cross-linked gelatin hydrogel micro particles were encapsulated with fibroblast growth factor 2 (FGF-2) and mixed with collagen sponge pieces	Detection of expression of *DSPP* (Dentin Sialophosphoprotein)	Regulated FGF2 release from gelatin hydrogels resulted in the production of dentin-like particulates with dentin defects above exposed pulp.	[133]
Fibrin	Incorporation of clindamycin loaded Poly (D, L) Lactic Acid nanoparticles (CLIN-loaded PLA NPs).	Cell viability and antimicrobial assay	Fibrin hydrogels incorporating CLIN-loaded PLA NPs reduced bacterial colonization and had an antimicrobial property towards *E. faecalis*. In cellularized hydrogels, DPSC survival and type I collagen production were comparable to the unmodified groups.	[134]
Silk Fibroin	Silk Fibroin/Cellulose Hydrogel	Cell viability	The hydrogels promote MC3T3 cell development into osteoblasts and are predicted to be a promising matrix for osteogenesis.	[135]

**Table 7 pharmaceuticals-16-00702-t007:** Hydrogel in synthetic materials.

Hydrogel	Delivery Vehicle	Experiment Design	Outcome	Reference
Polylactic polyglycolic acid–polyethylene glycol (PLGA-PEG)		Clinical Trial	A biological strategy may create a conditions favorable to therapeutic regeneration of dental and paradental tissues.	[148]
PLGA	Lactoferrin and substance P in a chitin/PLGA-CaSO_4_ hydrogel	Clavarial rat defect	In mice, clavarial bone regeneration was enhanced compared to controls.	[149]
PEG	PEG–maleate–citrate (PEGMC) (45% *w*/*v*), acrylic acid (AA) cross linker (5% *w*/*v*), 2.20-Azobis (2-methylpropionamidine)dihydrochloride (AAPH) photo-initiator (0.1% *w*/*v*),	Cell viability		[150]
PEG	A Tetra-PEG Hydrogel Based Aspirin Sustained Release System	In vitro and in vivo analyses	When periodontal ligament stem cells (PDLSCs) were co-incubated with hydrogel materials, in vitro tests revealed that cell growth was somewhat aided and osteogenic development was significantly enhanced. Furthermore, an in vivo investigation revealed that the aspirin controlled release approach greatly aided PDLSCs-mediated bone defect healing.	[151]
Poly-Nisopropylacrylamide (NIPAAm)	NIPAAm cross-linked by PEG-DMA	DSPP in the outer cell layer.	DPSCs in the construct’s outermost surface developed into odontoblast-like cells, whilst DPSCs in the inner layer remained stem cells. In vivo, blood vessel-rich pulp-like tissues were created.	[152]

**Table 8 pharmaceuticals-16-00702-t008:** Hydrogel materials applications.

Material		Application	Outcomes	Reference
Alginate-Matrigel hydrogel encapsulated with bioactive glass micro particles		In vitro (human dental pulp MSCs)	Despite a decrease in elasticity due to the incorporation of bioactive glass microparticles, the incorporation of Matrigel in the hydrogel combination promotes MSC osteogenic differentiation.	[12]
3D-printed heparin-collagen network enclosing MSCs, reinforced with -TCP nanoscale framework, and complexed with human bone morphogenetic protein type 2 (rhBMP-2)		In vitro (human dental pulp MSCs); In vivo (rat dorsum defects)	In vitro: the capability of heparin-conjugated collagen matrix to retain rhBMP-2 bioactivity and improve MSC survival and osteogenic growth. MSCs’ osteogenic differentiation capacity and the formation of ectopic osteogenesis in vivo	[159]
Bacterial cellulose encapsulated with bone morphogenetic protein type 2 (BMP-2)		In vivo (frontal sinus lift rabbit model)	Bacterial cellulose has demonstrated great biocompatibility. Bacterial cellulose, when combined with BMP-2, aided bone repair while also acting as a barrier membrane and a drug release prolonger, as indicated by histological and immunohistochemical tests.	[160]
Chitosan-Gelatin hydrogel incorporating nanodimensional bioactive glass particles		Human dental pulp MSCs in vitro; rat femoral deformities in vivo	Biocompatibility and ability to generate bone-like apatite crystallization in vitro In vivo, the chitosan-gelatin hydrogel containing 5% bioactive glass nanostructures generated the highest bone regeneration outcomes.	[161]
Composite bisphosphonate-linked hyaluronic acid-calcium phosphate hydrogel		In vivo (sinus lift rabbit model)	In a histomorphometric analysis, the synthetic granular calcium phosphate material and deproteinized cow mineral xenograft stimulated more bone repair than the hyaluronic acid-calcium phosphate hydrogel.	[162]
Gelatin-coated β-tricalcium phosphate (βTCP) scaffolds with rhBMP-2-loaded chitosan nanoparticles delivery system		In vitro (human buccal fat pad MSCs)	Gelatin-coated TCP scaffolding with rhBMP-2-loaded chitosan nanoparticles promoted cell survival and adhesion while progressively releasing rhBMP-2 at a therapeutic dosage that allowed MSCs to develop into osteoblasts.	[163]
Alginate-gelatin methacrylate (GelMA) hydrogel		In vitro (human gingival MSCs and human bone marrow MSCs)	Because of the hydrogel’s reduced flexibility, the addition of GelMA to alginate impairs the hydrogel osteogenic development beginning of encapsulated MSCs. The biological characteristics of alginateGelMA, as well as the existence of inductive cues, govern MSC differentiation into osteoblasts.	[164]
Crosslinked pNIPAM-co-DMAc hydrogel loaded with hydroxyapatite nanoparticles		In vitro (commercial human MSCs); In vivo (rat femur defects)	Commercial human MSCs’ capacity to drive osteogenic development in vitro; in vivo: biocompatibility, capacity to combine with surrounding structures, and enhanced accumulation of early indicators of bone regeneration.	[165,166]
3D-bioprinted biphasic osteon-like framework comprising human mesenchymal stem cells (hMSCs) and human umbilical vein endothelial cells (HUVECs) enclosed in a fibrin-polycaprolactone hydrogel		In vitro (commercial hMSCs and HUVECs) In vivo (rat cranial bone defects)	in vitro; Significant increase in transcription of angiogenic biomarkers In vivo: histological analysis of explanted biomaterials demonstrated a boost in the quantity of blood vessels per square meter (the capacity to stimulate angiogenesis) in the three-dimensional bioprinting osteon-like framework.	[167]
Sodium alginate/hydroxyethylcellulose/ hydroxyapatite composite	Semi-synthetic	In vitro (commercial human MSCs); In vivo (rat femur defects)	The capacity of the hydrogel composite scaffolds to support hMSC cell survival and proliferation in vitro. Histological studies demonstrated neo-osteogenesis to heal the damaged sites 6 weeks following scaffold placement.	[168,169]
3D polyvinyl alcohol-tetraethylorthosilicatealginate-calcium oxide biocomposite cryogels	Semi-synthetic	In vivo (rat cranial bone defects)	The bone defect is allowed to repair during a 4-week period while its components are recirculated from the defect area. Osteoblastic function at the injured area, with a 2 to 4 week surge in development towards the osteoblastic lineage and osteoblast maturation.	[170]
Triblock poly(ethylene glycol)-poly (*ε*-caprolactone)-poly(ethylene glycol) copolymer, collagen and nanohydroxyapatite	Semi-synthetic, injectable	In vivo (rabbit calvarial bone defects)	After 4, 12, and 20 weeks, bone regeneration was evaluated. The creation of new bone tissue from the border of defects and the surface of native bone towards the center was established by radiological and pathological investigations. Non-loading defects have a great potentiality for correction via minimally invasive surgical procedures.	[171]

## Data Availability

Data can be provided on request to corresponding authors.

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
