# Peer review of "New Challenges and Prospective Applications of Three-Dimensional Bioactive Polymeric Hydrogels in Oral and Craniofacial Tissue Engineering: A Narrative Review"

_pharmaceuticals, 2023, doi:10.3390/ph16050702_

Round 1
Reviewer 1 Report
I think it is a very interesting article that is written from
clear and very educational way for those who have the interest to know
about 3D scaffolding and its applications. The happy ones, it's a
job very well done.
Author Response
REVIEWER 1
I think it is a very interesting article that is written from clear and very educational way for those who have the interest to know about 3D scaffolding and its applications. The happy ones, it's a job very well done.
Response: We want to thank reviewer 1 for his/ her encouraging comment.
Reviewer 2 Report
1. The whole article needs to be condensed into a clear narrative closely related to the thesis of "Dental and Osseous Tissue Regeneration". A clear logic line focusing on this thesis needs to be presented throughout all subpoints and references. The author should consider either to adjust the title or the chapter content.
a. A lot paragraphs only contain one sentences (e.g: Line 540-556), which need to be combined or reorgnized.
b. Too much unrelated work is introduced until Chapter 9 (e.g: Ref 90-101 not related to the thesis of dental and osseous tissue regeneration), making the article too lengthy. Thoes unnecessary paragraphs should be condensed or deleted.
c. The advantage of using injectable biomaterials in dental and osseous tissue regeneration work should be discussed.
d. Figure 2 is not essential or helpful for readers to understand the article.
2. Some confusing expression in the article should be improved.
e.g: Line 48-49
Line 67 "are needed very once"
3. The hydrogel is normally treated as soft material which is mechanically different from bone/dental tissue. The essentiality of increasing mechanical characteristics is mentioned in the conclusion part, however, detailed discussion would be preferred to being concluded in main chapter.
Author Response
RESPONSE TO REVIEWER 2 ;
We want to show gratitude to reviewer 2 for his inspiring comments. We have revised the whole manuscript according to his valuable suggestions.
Comments:
- The whole article needs to be condensed into a clear narrative closely related to the thesis of "Dental and Osseous Tissue Regeneration". A clear logic line focusing on this thesis needs to be presented throughout all subpoints and references. The author should consider either to adjust the title or the chapter content.
Response: We have replaced the words in the title“Dental, and Osseous Tissue Regeneration”, with “Oral, and Craniofacial Tissue Engineering ”.
- A lot paragraphs only contain one sentences (e.g: Line 540-556), which need to be combined or reorgnized.
Response: A) Thank you for your constructive comments. We have tried to improve our revised manuscript according to your valuable suggestions. We have removed “heparin, and chondroitin sulfate “small paragraphs line “540-562” in the old manuscript.
We have removed un related references “90-101” in the old manuscript, and, and replaced them with related ones.
We have added, and modified paragraphs in the revised manuscript in lines “47Ù€51”, “103Ù€104”, replaced words “dental, and osseous engineering” in line 131 in the old manuscript with “oral, and craniofacial ” in line 116 in the revised manuscript.
We have replaced the words “preferably be bioactive and biodegradable” in line 110 in the old manuscript with “has to possess sufficient bioactivity, and biodegradability” in line “103-104” in the revised manuscript,
We have replaced the words “Synthetic polymers with synthetic polymer” in table 1 in the old manuscript with “Two or more artificial scaffolds” in the revised manuscript.
We have changed the wrods in lines “155-158”, in the old manuscript with words in lines “139-141” in the revised manuscript.
We have channged the sentence “Hydrogels made from natural polymers” in line “176-177” in the old manuscript to “Hydrogels made from natural polymers” in line “155-156” in the revised manuscript.
We have changed the word non toxic in line 177 in the old manuscript to safe in line 156 in the revised manuscript.
WE have changed the sentence “dangerous to host cells and tissues if not properly removed “ in line 183 in the old manuscript with the sentence “may be cytotoxic if not properly eliminatd” in line 161 in the revised manuscript.
WE have changed the word “structural integrity” in line 207 in the old manuscript to word “Mechanical” in line 180 in the revised manuscript.
We have changed the word “density” in line 249 in the old manuscript to the word “capacity” in line 219 in the revised manuscript.
WE have replaced the paragraph in line “264-274” in the old manuscript to another one in lines “231-236” in the revised manuscript.
We have replaced reference 88 in line 275 in the old manuscript with reference 48,49 in line 240 in the revised manuscript.
WE have removed some names from table 3 “Casein, Zein, Gliadin, Legumin, Elastin”
We have added a paragraph in line “252-257” in the revised manuscript.
We have removed sentences from line “316-324” in the old manuscript and added new ones in lines “266-277” in the revised manuscript.
We have replaced the sentence has various benefits as a biomaterial, including biocompatibility and non-immunogenicity in line “325-326” in the old manuscript to is biocompatible, and non-immunogenic in line “261-262” in the revised manuscript.
We have replaced the word “cartilage” in line 332 in the old manuscript with the word “body “in line 279 in the revised manuscript.
We have removed sentences from line “332-357” in the old manuscript.
We have add a paragraph in line “282-285” in the revised manuscript.
We have added sentences in line “393-395” in the revised paragraph.
WE have replaced the word “availability” in line 391 in the old manuscript with the word “abundance” in line 303 in the revised manuscript.
We have added a paragraph in line “312-315” in the revised manuscript.
We have removed sentences in line “444-449” in the old manuscript.
WE have replaced the word “extensively” in line 451 in the old manuscript with the word “widely” in line 318 in the revised manuscript.
We have added a paragraph in line “323-328” in the revised manuscript.
We have removed sentences in lines “469-490” in the old manuscript.
We have added the sentence with a reference “dental tissue regeneration [74].” in line “332-333” in the revised manuscript.
We have removed the sentences in lines “494-501” in the old manuscript.
We have added a paragraph in line “337-342” in the revised manuscript.
We have added “358-361” in the revised manuscript.
We have replaced the sentence “a number of advantages” in line 530 in the old manuscript with th sentences “outstanding biological features” in line 363 in the revised manuscript.
We have replaced the sentence “their use as potential biomaterials in a variety of different applications.” in lines “531-532” with the sentence “their popularity in numerous purposes” in line 364 in the revised manuscript.
We have added a new paragraph in line “369-371” in the revised manuscript .
We have removed the item “Experimental design ” from table 5 in the old manuscript.
We have added new sentences in line “401-414” in the revised manuscript.
WE have added new sentences in lines “428-436” in the revised manuscript.
We have added new sentences in “451-463” in the revised manuscript.
We have removed sentences in lines “628-631” in the old manuscript.
We have replaced paragraphs in line “649-673” in the old manuscript with paragraphs in line “466-486” in the revised manuscript.
WE have removed sentences from line “751-756” in the old manuscript.
We have added new sentences in line “519-523” in the revised manuscript.
We have added new sentences in line “534-544” in the revised manuscript.
We have added new sentences in line “556-562“ in the revised manuscript.
We have replaced the word “addition” in table 8 in the old manuscript with the word “incorporation” I the revised manuscript.
We have replaced the word “capacity” I table 8 in the old manuscript with the word “capability” in the revised manuscript.
We have replaced the sentence “greatest new bone “ in the old manuscript with the sentence “Highest bone regeneration outcomes” in the revised manuscript.
We have replaced the sentence n line “801-802” in the old manuscript to “composed of a single component” in line 605 in the revised manuscript.
- Too much unrelated work is introduced until Chapter 9 (e.g: Ref 90-101 not related to the thesis of dental and osseous tissue regeneration), making the article too lengthy. Those unnecessary paragraphs should be condensed or deleted.
Response: We have revised and added the “Injectable hydrogels” in the revised manuscript line “802-816”.
- The advantage of using injectable biomaterials in dental and osseous tissue regeneration work should be discussed.
Response: WE have replaced the sentences in lines “807-809” in the old manuscript to “are formed of two or more different types” in the revised manuscript.
- Figure 2 is not essential or helpful for readers to understand the article.
Response: We have modified figure 2.
- Some confusing expression in the article should be improved.
e.g: Line 48-49
Line 67 "are needed very once"
Response: We have removed the confusing expressions in the old manuscript line “47-48” to new one line “47-51”, and words in line “67” in the old manuscript to “are need for ”.
- The hydrogel is normally treated as soft material which is mechanically different from bone/dental tissue. The essentiality of increasing mechanical characteristics is mentioned in the conclusion part, however, detailed discussion would be preferred to being concluded in main chapter.
Response: We have added a more detailed discussion on the importance of the mechanical properties, and their enhancement in lines “189-198” in the revised manuscript.

Reviewer 3 Report
I am pleased to review the article "Three Dimensional Bioactive Polymeric Hydrogel Scaffolds for 2 Dental and Osseous Tissue Regeneration Engineering: New 3 Challenges and Prospective Applications: A Narrative Review " Article is well-written with appropriate references. The content of this article is very important in the field of regenerative medicine in both clinical application and research & development application.
Author Response
RESPONSE TO REVIEWER 3: We want to thank reviewer 3 for his/ her positive evaluation of our manuscript.